# Why Are the Proportions of In-Vitro Fertilisation Interventions for Same Sex Female Couples Increasing?

**DOI:** 10.3390/healthcare9121657

**Published:** 2021-11-30

**Authors:** Catherine Meads, Laura-Rose Thorogood, Katy Lindemann, Susan Bewley

**Affiliations:** 1Faculty of Health, Education, Medicine and Social Care, Anglia Ruskin University, Cambridge CB1 1PT, UK; 2LGBT Mummies Tribe, 3rd Floor, 207 Regent Street, London W1B 3HH, UK; contact@thelgbtmummiestribe.com; 3Uber Barrens Club, London, UK; katy@uberbarrens.club; 4Department of Women and Children’s Health, School of Life Course and Population Sciences, Faculty of Life Sciences & Medicine, King’s College London, London SE1 7EH, UK; susan.bewley@kcl.ac.uk

**Keywords:** same sex female couples, lesbians, bisexual women, IVF, donor insemination, pregnancy choices

## Abstract

Same-sex female couples who wish to become pregnant can choose donor insemination or in-vitro fertilization (IVF)—a technique intended for infertile women. In general, women in same-sex female partnerships are no more likely to be infertile than those in opposite sex partnerships. This article investigates data available from the Government Regulator of UK fertility clinics—the Human Fertilization and Embryology Authority, which is the only data available worldwide on same-sex female couples and their fertility choices. IVF is increasing both in absolute numbers and relative proportions year on year in the UK, compared to licensed donor insemination for same-sex female couples. As IVF has greater human and financial costs than donor insemination, policies should not encourage it as the first choice for fertile women requiring sperm. Commercial transactions are taking place where fertile lesbians receive cut price, and arguably unnecessary, IVF intervention in exchange for selling their eggs to be used for other infertile customers. If women are not told about the efficacy of fresh vs. frozen semen, and the risks of egg ‘sharing’ or intra-couple donation, exploitation becomes possible.

## 1. Introduction

A woman in a same-sex couple who wishes to have a baby will need to use an egg and sperm from somewhere. The options in the UK are:Her own egg, and fresh sperm from a known male or unknown one found online, using an informal, unregulated arrangementHer own egg, and sperm (usually frozen) from a formal, regulated healthcare operator (National Health Service (NHS) or private, including located abroad)Her partner’s egg, and sperm (usually frozen) from a formal, regulated healthcare operator (NHS or private, including located abroad)A third-party donated egg, and sperm (usually frozen) from a formal regulated healthcare operator (NHS or private, including located abroad)

All options contain uncertainties.

Option 1 may be viewed as preferable by some due to being cheaper and less invasive. It uses home insemination and fresh semen (which has greater success than frozen [1], relationships rely on trust, and the child usually knows its origins, but few resources exist for General Practitioners (GPs) or clinics to discuss the options [2]). By contrast, options 2–4 use formal, regulated healthcare operators and have the qualities of a rigorous assessment process, medical advice and involvement, controlled environments, and contractual legality, thus entailing greater cost.

Option 2 can be done in two ways. The first (2a) is where washed sperm is inserted into the genetic and gestational mother’s uterus using an intra-uterine cannula, a procedure known as donor insemination–intra-uterine insemination (DI-IUI). This procedure can be done with or without ovarian stimulation. If frozen sperm is used, intracytoplasmic sperm injection (ICSI) is more likely to be needed. The side effects of DI-IUI without stimulation are minimal; some women may experience stomach cramps similar to period pains. With stimulated cycles (timed ovulation induction), side effects include bruising, bloating, constipation, breast tenderness, and moodiness. This option is also associated with an increased risk of multiple pregnancy. Option 2a costs around £800–£1300 per cycle.

Alternatively (2b), in-vitro fertilisation (IVF) can be used. In IVF, ovulation is stimulated and controlled by drugs, ova are extracted under transvaginal ultrasound control from the mature follicles growing in the woman’s ovaries and combined with donated sperm in a laboratory dish, or by using ICSI. The resulting embryo(s) are then transplanted back into her uterus. Spare embryos may also be frozen to be used in the future, or donated to medical research or another couple. IVF is not directly comparable to DI, partly because hyperstimulation is used to increase the numbers of eggs compared to natural or timed cycles in DI, and partly because of occasional multiple embryo transfers. With IVF there is control over the number of embryos transferred, and spare embryos might be cryopreserved. Option 2b costs around £5000–£7000 per cycle depending on protocol and medication used. Women may be less likely to undergo as many repeat IVFs as DIs given that it is more successful but also more costly.

The side effects and complications of IVF, although decreasing in frequency with more modern protocols, include hot flushes, allergic reactions, and complications of the drugs, including ovarian hyperstimulation syndrome (OHSS), depression or irritability, vaginal bleeding, blood in urine, bloating, breast tenderness, bruising from injections, cramping, headaches, restlessness, pelvic pain, and constipation. These can persist for years in some women, for example if the IVF triggers depression [3], or be lifelong for children, for example if they suffer prematurity-related morbidities after IVF [4] that happen with single embryo transfer but especially with multiple pregnancy [5]. Adverse pregnancy outcomes are also increased, especially relating to multiple births. Severe OHSS occurs in approximately 1.4 % of all cycles [6] and cases have appeared in the past in the Confidential Enquiry into Maternal Deaths [7]. The rate of a wide variety of perinatal adverse events following IVF side effects is around double that of spontaneous conceptions [4,8]. Longer term, there is uncertainty about an association between IVF and borderline ovarian tumors [9] and endometrial tumors [10].

In option 3, IVF is also used, but one woman’s embryo is transferred into her partner’s uterus after she too has been primed with drugs. A woman who undergoes egg collection completes HFEA (Human Fertilisation and Embryology Authority) paperwork as a donor, and the woman who undergoes embryo transfer completes paperwork as a patient undergoing double donor IVF. The substantially greater physical risks of any pregnancy are taken by the embryo recipient. In addition, obstetric complications may arise because of increased genetic incompatibility between gestational mother and fetus who are now 100% (rather than 50%) unalike, in particular diseases such as preeclampsia with its almost five-fold increase in risk [11]. This procedure costs around £8000–£9000 per cycle plus extras.

Option 4 is used where the woman herself is infertile and Option 3 is not possible or desired. Around one in seven UK couples may have difficulty conceiving [10]. When age differences are taken into account, there are reportedly few differences in fertility rates in lesbians compared to heterosexual women [12]. A recent systematic review found that pregnancy rates were similar or higher in lesbians than heterosexual women after DI [13].

In an ideal world, we would have information on: all of the routes taken by same-sex female couples; the success rates per procedure and per woman; multiple birth rates; and side effects of each treatment, especially the regulated options 2–4. We would also have information on trans and non-binary people and their routes to medically-assisted pregnancy.

The Human Fertilisation and Embryology Authority (HFEA) is the UK Government’s regulatory body responsible for ensuring fertility clinics comply with UK law. It collects data from all UK fertility clinics and publishes an annual report and accompanying data sheet. The HFEA found little difference in birth rates between same-sex and opposite-sex couples after donor insemination (DI) [14]. The HFEA does not collect data separately on trans and non-binary people and their interactions with fertility clinics, these are included in the generic data.

## 2. Materials and Methods

This project is secondary data analysis of freely available data on fertility trends in the UK. The data is provided by the HFEA at their website [14,15,16]. Data were downloaded from their most recent spreadsheet, relating to data collected by them up to 2019. We analyzed these data in an Excel spreadsheet and percentages were derived. Members of a support group for LGBT+ women & people worldwide on the path to motherhood or parenthood were consulted by the Lead Contact (L-R T) asking about their recent experiences of UK fertility clinics as an informal service evaluation and for hypothesis generation. Comments contributed were constructed into a range of typical, illustrative stories.

## 3. Results

### 3.1. Fertility Trends

For option 1 above, there is no information regarding pregnancy rates for same-sex female couples using private arrangements with men who donate fresh sperm. Fresh sperm is deposited by the cervix whereas frozen clinic sperm is thawed then instilled into the uterus via a catheter. There is no clear evidence as to which route for frozen sperm has better success rates [17]. Long ago, fresh DI was shown to be three-fold better in achieving pregnancy than frozen DI for infertile women used as their own controls [3]. Logically, using fresh, recently produced semen at the time of ovulation would also be expected to have success rates similar to a natural background in fertile (or untested) women, and better success rates per intervention than in current HFEA data of 13.8% (unstimulated) and 15.2% (stimulated) with frozen sperm (option 2a), and 27.8% with IVF (options 2b—patient’s egg, donor sperm) and 36.6% with options 3–4 (donor egg and donor sperm) [15].

The numbers of same-sex female couples using DI and IVF are increasing overall. Fewer same sex female couples are now receiving DI-IUI than IVF (see Table 1). The HFEA report from 2019 [16] included data up to 2017 and stated that there is “an upward trend in the use of IVF by patients in same-sex partnerships [and also for single women with no partner] and clinicians recommending trying IVF before DI”. Their Family Formations Report [14] states that “Historically, most patients in same-sex relationships… have used DI, as these patients are likely seeking treatment to access donor sperm rather than for infertility reasons.” It also states that “almost 60% of patients in female same-sex relationships seeking fertility treatment started IVF without any prior DI cycles in 2018”. It notes that “The increased use of IVF may relate to the higher birth rate of IVF compared to DI and to the added cost of undergoing multiple rounds of DI cycles to achieve a birth.”

The HFEA datasheet [15] gives the proportions of patients experiencing adverse events from IVF and DI treatment. These include miscarriages, ectopic pregnancies, heterotopic, biochemical, and molar. In 2019, with IVF, of 22,866 pregnancies there were 2657 miscarriages (11.7%), 235 ectopic pregnancies (1.0%), and 4319 biochemical adverse events (18.9%); with DI of 987 pregnancies, there were 103 miscarriages (10.4%), 11 ectopic pregnancies (1.1%), and 109 biochemical adverse events (11.0%).

The average age of IVF is now lower in same-sex than opposite-sex couples (34.9 vs. 35.6 years). The two types of IVF within a same-sex female couple, when the egg from one woman is used within her own or her partner’s uterus, are not distinguished in the HFEA dataset because the donor egg could come from a partner or another donor. The proportion of same-sex female couple procedures that may be intra-partner egg donation is unclear.

A review of statistics from other countries yielded no relevant information. The Canadian Assisted Reproductive Technologies Register (CARTR) report based on data submitted by clinics to the CARTR Plus database mentions reasons for treatment including “no male partner” but no further clarification [18]. The USA National Center for Chronic Disease Prevention and Health Promotion Assisted Reproduction Success Report 2017 does not include same-sex female couples as a reason for using assisted reproduction techniques (ART) [19] and there do not appear to be any statistics about this on their website. The European Society of Human Reproduction and Embryology core dataset does not include partner sex so cannot be used to find ART data for same-sex female couples [20]. So, the data presented here may be the only information available worldwide on rates of IVF and DI in same-sex female couples.

### 3.2. Personal Experiences

The following three reconstructed stories summarize scenarios typical of a range of feedback to the LGBT Mummies Tribe from same-sex female couples around their experiences of using fertility clinics and using IVF rather than DI-IUI.

Couple 1: They said they felt “*steered*” towards IVF rather than DI-IUI, IUI was hardly spoken about and dismissed, and that IVF was the quickest & best option, even though they had no fertility issues.

Couple 2. They said they attended their clinic with the idea of IVF in mind due to wanting to achieve a pregnancy quickly. The clinic gave them their options of both DI-IUI & IVF. They said they felt that the clinic covered the risks of both, and success rates equally. They decided to go for IVF because they were aware of DI-IUI taking longer, and felt that their clinic was helpful in regards to giving them options around the different routes to take. They were happy with the service they received.

Couple 3: The clinic did not talk about DI-IUI at all, they only sold IVF at the open evening. The couple then wanted to do ‘reciprocal’ IVF and the clinic advised they could ‘egg share’ and provide eggs for someone else. They did not receive any guidance or information relating to the risks of egg sharing. They were told that it was a ‘package deal’, so if they participated, it meant they got treatment for £500. Later, they received multiple bills for hidden fees. ICSI was performed without their knowledge. The couple fought this for months before the charge was removed. With the second round of egg sharing, the clinic refused to do a fresh transfer and said frozen was better due to her having the complication of OHSS. 

“*We felt lied to, and that I was just being exploited and harvested for heterosexuals who needed the eggs. The fact that if I didn’t produce 12 eggs or more, meant we had to pay the full amount instead of the £500 for the treatment, meant additional stress on us, & at no point were we told of any risks with egg sharing—we did egg sharing 3 times*.” 

## 4. Discussion

UK National Institute of Health and Care Excellence (NICE) guidelines recommend that women aged under 40 years who have not conceived after 2 years of regular unprotected heterosexual intercourse or 12 cycles of artificial insemination (where six or more are by DI-IUI) should be offered IVF [21]. Like most fertility services, this is affected by local commissioning arrangements. Given this requirement, it is unlikely that same-sex female couples will be referred to an NHS clinic for a new appointment.

So, if a same-sex female couple attends a private fertility clinic as a new appointment, they may be presented with the option to decide between paying for donor sperm (Option 2a) or to consider the much more invasive and costly IVF (Options 2b-4). They should be informed of the risks and side effects of IVF regarding Option 2b or Option 3, or whether they need to use Option 4 because of infertility. If they were unaware of the efficacy of home insemination (Option 1), they might never have needed any medical involvement in the first place. However, the couple should also be advised to seek legal advice regarding what happens in the event of breakdown of trust. With home insemination, the sperm donor has legal right to access the child and can be pursued for maintenance, unless legal procedures are completed correctly before conception takes place, so that the requirements are met for legal parenthood to be established for the recipient and partner at the point of conception. Fertility clinics have established processes to complete this for their clients.

Additionally, the legal change in April 2005 requiring sperm donors to provide their personal details in order for the child to know its origins, has meant that same-sex female couples purchasing sperm can now only do this via a regulated clinic. As couples cannot legally purchase sperm from abroad to be delivered direct to their home, this may have several consequences such as the financial pressure on low-waged lesbians to opt for a 1-night stand strategy; increased use of IVF clinics abroad; and same-sex female couples being persuaded by clinics to opt for more invasive and expensive treatments, possibly with discounted terms and arrangements. Possibly because of this legal change requiring sperm donors to provide personal details, donor sperm is in short supply and is relatively expensive (£1000 per sample). This is another factor that may be pushing same-sex female couples towards IVF; multiple samples of sperm may be needed to complete the several cycles of DI possibly required to achieve pregnancy, whereas fewer may be needed for IVF. Another possibility is that a woman in a same-sex couple may object to having sperm in her vagina, although there is no evidence about this. This would be expected to remain relatively constant so would not explain the rise in IVF proportions.

Reasons why the relative proportions of DI vs IVF treatments are gradually falling might include patient choice, legal changes, financial concerns, relative success rates, and advertising amongst others. It is unlikely that the explanation for same-sex female couples opting for IVF more than ever before is an increase in their underlying medical infertility over the 2 decades in question. Financially, couples have to make a choice between paying for several attempts at DI first because they are not always eligible for NHS treatment (as eligibility criteria vary across the country), or for one or more IVF cycles. DI has lower individual success rates per cycle than IVF but costs less per attempt. Per cycle, pregnancy is more likely to have higher live birth rates with IVF than DI, but finances may limit the number of attempts. Although there is no evidence regarding lesbians’ views, clinicians may consider that there will be accumulating stress after repeated IUI failures and counsel accordingly. They may be concerned (as may patients) about the time-to-pregnancy as a crucial quality indicator, and then put this into routine counselling discussions. There is a fine line between patient choice based on non-directive counselling and ‘doctor-led behavior’. A market is made up of buyers and sellers who both have influence. Part of the change may be driven by couples themselves, their preferences, or their own financial constraints, and not by the clinics. However, these customers rely on the descriptions of the products and their doctors’ beliefs and recommendations. 

There are higher risks and worse side effects with IVF than DI. If a woman is approaching 40 years old, if DI has failed, or if she is unwilling to undergo six cycles of DI first, then IVF may be appropriate. Women may be less likely to have as many repeat IVFs, given that it is more onerous and costly. The rise in IVF proportions may also be due to clinics promoting IVF as being much safer than in the past. For example, there are lower rates of OHSS. Doctors may be keen to offer pre-implantation genetic screening before pregnancy, in order to reassure couples about the presence of aneuploidy. Techniques of embryo cryopreservation (vitrification) have improved, allowing more future options, but also at a price. The age of people seeking parenthood is steadily increasing overall, and higher age reduces success rates, particularly for DI. However, the average age of same sex female couples in the HFEA dataset is gradually reducing, from 35.5 (SD 4.8) in 2000 to 34.8 (SD 4.5) in 2019, for women receiving IVF, and 34.4 (SD 4.5) in 2000 to 33.2 (SD 4.2) in 2019 for women receiving DI [15]. 

Couples sometimes select IVF for its possibility for intra-partner egg donation [12], which is often presented as a romantic way to conceive a baby within a female couple. The narrative is that both partners have some biological connection with their child, which is a powerful psychological, emotional, and social motivator. Websites frequently present an encouraging view of this so-called ‘reciprocal’ or ‘partner IVF’ process [22]. Medically, this is an intra-couple altruistic egg donation arrangement, involving medical third-parties and three different biological providers (of egg, sperm, and uterus), which carries higher obstetric risk to the non-genetically-related woman. There is an ethical issue for the doctor here, it is not a simple consumer choice.

In some clinics, a woman being hyperstimulated to donate eggs, including to her partner through IVF, can also give eggs away in order to get reduced fee treatment. This is referred to as “egg-sharing” and is presented as a reasonable commercial arrangement with fertility companies, who “harvest” eggs from fertile women in exchange for IVF. Unlike other forms of egg “bartering” or “selling” [23], clinics may be particularly interested in obtaining lesbians’ eggs, as it appears they achieve higher pregnancy rates for recipients, maybe unsurprisingly given that few donors are medically infertile. Only women aged under 35 can “egg share”. Thus, in order to obtain the associated financial discount, women averaging age 34 may feel they have insufficient time to try several DI cycles first before opting for IVF. Commercial pressures also exist in a profit-driven industry that has a severe shortage of eggs for heterosexual couples suffering largely with age-related or unexplained infertility. The (usually older and heterosexual) woman who is the egg-sharing recipient may also not have been told about the extra risks of an ovum recipient pregnancy and she too may suffer avoidable, iatrogenic complications. 

Overall, excess and unnecessary use of IVF may incur avoidable side effects and complications, including maternal death [24,25]. Fertility clinics effectively offset the full human and financial costs of IVF, as the NHS deals with the side effects and poor outcomes, including treatment of pre-eclampsia, multiple births, depression, and prematurity-related morbidities in the child. If the number of women using IVF is increasing, it will inevitably have a detrimental effect on women, children, and the NHS, not the fertility clinics, their staff, or shareholders.

## 5. Conclusions

Same-sex female couples who wish to have a baby and use formal fertility clinic services can choose between DI and IVF in the private sector. Both have advantages and disadvantages. The rates of DI relative to IVF interventions are falling, but the reasons why are not straightforward or clear. They may relate more to fertility clinic-led initiatives, patient choice, and commercial pressures rather than medical need. The clinics have a duty of care and are legally bound by the HFEA code of practice to provide patients with full information about the treatments available and attendant medical risks. Clinics must put patients first, rather than their own commercial interests. The HFEA should improve reporting, examine individual clinic practices, and ensure compliance.

## Figures and Tables

**Table 1 healthcare-09-01657-t001:** Numbers of patients and in-vitro fertilisation (IVF) and donor insemination (DI) treatment cycles with relative proportions.

Year of Treatment	Numbers of DI Treatment Cycles for Women with Female Partner	Numbers of Patients in Same-Sex Female Couples Receiving DI	Numbers of IVF Treatment Cycles for Women with Female Partner	Numbers of Patients in Same-Sex Female Couples Receiving IVF	Proportions of DI and IVF Intervention Cycles DI% IVF%		Proportions of DI and IVF Patients DI% IVF%	
2000	441	194	38	30	92.1	7.9	86.6	13.4
2001	585	214	46	32	92.7	7.3	87.0	13.0
2002	655	251	86	61	88.4	11.6	80.4	19.6
2003	720	291	85	66	89.4	10.6	81.5	18.5
2004	943	368	99	73	90.5	9.5	83.4	16.6
2005	873	376	151	121	85.3	14.7	75.7	24.3
2006	866	363	199	151	81.3	18.7	70.6	29.4
2007	708	322	261	184	73.1	26.9	63.6	36.4
2008	896	405	331	250	73.0	27.0	61.8	38.2
2009	984	469	489	352	66.8	33.2	57.1	42.9
2010	1091	559	588	437	65.0	35.0	56.1	43.9
2011	1310	641	789	579	62.4	37.6	52.5	47.5
2012	1484	743	943	694	61.1	38.9	51.7	48.3
2013	1533	789	1092	803	58.4	41.6	49.6	50.4
2014	1845	944	1331	986	58.1	41.9	48.9	51.1
2015	2065	1051	1461	1091	58.6	41.4	49.1	50.9
2016	2298	1125	1735	1249	57.0	43.0	47.4	52.6
2017	2517	1333	2044	1451	55.2	44.8	47.9	52.1
2018	2616	1346	2203	1574	54.3	45.7	46.1	53.9
2019	2514	1313	2435	1688	50.8	49.2	43.8	56.2

## Data Availability

Data available at https://www.hfea.gov.uk/.

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
