# Peer review of "Why Are the Proportions of In-Vitro Fertilisation Interventions for Same Sex Female Couples Increasing?"

_healthcare, 2021, doi:10.3390/healthcare9121657_

Round 1
Reviewer 1 Report
The work is interesting in particular for a reviewer from a country where the use of insemination and art in same-sex couples is not allowed. I just wonder if the tendency to use in vitro fertilization instead of insemination is not influenced by the lack of acceptance of sperm administration to the vagine . I see it in heterosexual couples where donor sperm need to be used. Lack of acceptance of the placement of sperm in vagine and acceptance transfer of embryos after fertilization with donor sperm.
Author Response
Peer reviewer 1
Comment |
Response |
Action |
The work is interesting in particular for a reviewer from a country where the use of insemination and art in same-sex couples is not allowed. I just wonder if the tendency to use in vitro fertilization instead of insemination is not influenced by the lack of acceptance of sperm administration to the vagine . I see it in heterosexual couples where donor sperm need to be used. Lack of acceptance of the placement of sperm in vagine and acceptance transfer of embryos after fertilization with donor sperm. |
Thank you. This is an interesting point. It appears the reviewer asks whether women will object to sperm in the vagina but not to a fertilized embryo in the uterus. Despite LRTs connection to several thousand LGBTmummies she has never heard this objection voiced. We have no evidence at the moment that this is an issue for same sex female couples. We don’t believe it would explain a changing trend, but have added it as a possibility. |
Addition at line 226-9 Another possibility is that a woman in a same sex couple may object to having sperm in her vagina, although there is no evidence about this. This would be expected to remain relatively constant so would not explain the rise in IVF proportions. |
Reviewer 2 Report
The manuscript is interesting and well written. Matters are sensibly discussed, and the literature is cited appropriately.
I wish to add some comments.
Obviously, the Authors' point of view is critical with respect to D-IVF. Central to their reasoning is the fact that the increasing rate of D-IVF cycles would depend mainly on pressure by private clinics. In other words, it is somehow given from granted that business would be the main leverage for such a trend. Whereas it is necessary to stigmatize behaviors (well documented) such as those reported for cases of Couple 1 and 3, which have little to do with good clinical practice, I believe that making hypotheses about potential "one night stands with unknown men" is going a little too far.
Furthermore, on one hand the Authors grant women the ability to weight advantages/disadvantages of IVF vs IUI, on the other hand they seem not to recognize the same ability as for deciding whether to share their oocytes (once informed correctly, of course). The same applies to couples that choose option 3 (page 2, line 81). A woman might deliberately accept an increased risk of pre-eclampsia, basing her choice on personal feelings and believes (again, when the info is provided correctly).
I would suggest alternative explanations for the increased proportion of D-IVF (which of course would not exclude those discussed in the manuscript).
First of all, IVF procedures have increased the level of safety over the years. Cycle segmentation, for instance, has brought the chances of severe hyperstimulation syndrome close to zero. Pre-implantation genetic screening (PGT-A) allows couples to know the presence of aneuploidies (so frequent in our species) before pregnancy. Furthermore, the greatly increased efficacy and efficiency of embryo cryopreservation (vitrification) allows for more pregnancies (singleton), years apart, after one single IVF/ICSI.
At one point the Authors refer to depression as a consequence of IVF, a problem that could last for years. But what about the accumulating stress after repeated IUI failures? One major issue in the field of assisted reproduction is the time-to-pregnancy. This variable is today considered a crucial quality indicator, and it is currently part of the discussion when counseling couples.
Finally, the age of people seeking parenthood is steadily increasing. We all know that this has dramatic consequences on the success rate, in particular for IUI, when only one oocyte is available for a potential pregnancy
Author Response
Peer reviewer 2
Comment |
Response |
Action |
The manuscript is interesting and well written. Matters are sensibly discussed, and the literature is cited appropriately |
We thank the peer reviewer |
- |
I wish to add some comments. Obviously, the Authors' point of view is critical with respect to D-IVF. Central to their reasoning is the fact that the increasing rate of D-IVF cycles would depend mainly on pressure by private clinics. In other words, it is somehow given from granted that business would be the main leverage for such a trend. |
We agree that a market needs both a buyer and a seller who both have influence, and we have already acknowledged (at LINE 234-7) that part of the change may be driven by couples themselves, by their preferences or their own financial constraints and not by the clinics. However, the customers are very reliant on the descriptions of the products and their doctors’ beliefs and recommendations -If we haven’t made this clear we should make it clearer |
Added at line 243-6 A market is made up of buyers and sellers who both have influence. Part of the change may be driven by couples themselves, their preferences or their own financial constraints, and not by the clinics. However, these customers rely on the descriptions of the products. and their doctors’ beliefs and recommendations |
Whereas it is necessary to stigmatize behaviors (well documented) such as those reported for cases of Couple 1 and 3, which have little to do with good clinical practice, I believe that making hypotheses about potential "one night stands with unknown men" is going a little too far. |
We agree that this may be hypothesis and is not supported by evidence in the literature. LRT is connected to several thousand LGBTmummies and is not aware of any/many lesbians who have deliberately sought one night stands with previously unknown and untraceable men in order to get pregnant. It is a ‘technical possibility’. Nevertheless, we do not believe we do, nor should, stigmatise women who would do this although the peer reviewer appears to ‘disapprove’ in some way. We don’t have to emphasise this matter so have toned down the comment |
We have removed the words ‘unknown men’ from line 219 as that is implied in the ‘one night stand’ phrase. |
Furthermore, on one hand the Authors grant women the ability to weight advantages/ disadvantages of IVF vs IUI, on the other hand they seem not to recognize the same ability as for deciding whether to share their oocytes (once informed correctly, of course). The same applies to couples that choose option 3 (page 2, line 81). A woman might deliberately accept an increased risk of pre-eclampsia, basing her choice on personal feelings and believes (again, when the info is provided correctly). |
We disagree and apologise for not being sufficiently clear in the text. To us, this issue is: (1) whether couples are really getting sufficient information from the clinic they attend in order to give informed consent for these options; and (2) whether a truly voluntary decision can be made when there is no ‘choice’ - say ‘egg sharing or nothing’? We think that women are not being given sufficient information; IVF clinic evenings, consultations and consent forms do not contain this information. Additionally, there would be an ethical issue for the doctor offering an option that increases the risk of a life-threatening disorder. |
Added at line 267. There is an ethical issue for the doctor here, it’s not a simple consumer choice.
|
I would suggest alternative explanations for the increased proportion of D-IVF (which of course would not exclude those discussed in the manuscript). First of all, IVF procedures have increased the level of safety over the years. Cycle segmentation, for instance, has brought the chances of severe hyperstimulation syndrome close to zero. Pre-implantation genetic screening (PGT-A) allows couples to know the presence of aneuploidies (so frequent in our species) before pregnancy. Furthermore, the greatly increased efficacy and efficiency of embryo cryopreservation (vitrification) allows for more pregnancies (singleton), years apart, after one single IVF/ICSI. |
We thank the peer reviewer for these points. We agree that lowering of OHSS is good, but it is not a reason to choose IVF. We disagree that PGS is a boon, given that aneuploidy is relatively rare by the end of pregnancy, and that RCTs have shown that its application (sold in order to ‘reassure’ about abnormalities) results in lower pregnancy rates. We agree that cryopreservation increases options. |
Added at lines 250-55 The rise in IVF proportions may also be due to clinics promoting IVF as being much safer than in the past. For example, there are lower rates of OHSS. Doctors may be keen to offer pre-implantation genetic screening before pregnancy, in order to reassure couples about the presence of aneuploidy. Techniques of embryo cryopreservation (vitrification) have improved, allowing more future options, but also at a price. |
At one point the Authors refer to depression as a consequence of IVF, a problem that could last for years. But what about the accumulating stress after repeated IUI failures? One major issue in the field of assisted reproduction is the time-to-pregnancy. This variable is today considered a crucial quality indicator, and it is currently part of the discussion when counseling couples. |
We thank the reviewer for this point. Clinics may well value time to pregnancy most, but we don’t know that is true for same sex female couples. This does not work as an explanation for why this might be changing. The peer reviewer’s explanation (‘currently part of the discussion’) does strongly suggest that patients’ behaviour and choices will be heavily influenced and ‘doctor-led’. |
Added at lines 238-42: Although there is no evidence regarding lesbians’ views, clinicians may consider that there will be accumulating stress after repeated IUI failures and counsel accordingly. They may be concerned (as may patients) about the time-to-pregnancy as a crucial quality indicator, and then put this into routine counselling discussions. There is a fine line between patient choice based on non-directive counselling and ‘doctor-led behaviour’. |
Finally, the age of people seeking parenthood is steadily increasing. We all know that this has dramatic consequences on the success rate, in particular for IUI, when only one oocyte is available for a potential pregnancy |
We agree. Except that’s not true for same sex female couples where the average age is decreasing. |
Added to lines 255-9: The age of people seeking parenthood is steadily increasing overall, and higher age reduces success rates, particularly for DI. However, the average age of same sex female couples in the HFEA dataset is gradually reducing, from 35.5 (SD 4.8) in 2000 to 34.8 (SD 4.5) in 2019, for women receiving IVF, and 34.4 (SD4.5) in 2000 to 33.2 (SD 4.2) in 2019 for women receiving DI[16]. |